environmental chemistry

heavy metals, soil, Shanxi

**Author for correspondence:**
Lihong Li
e-mail: lilh@jzxy.edu.cn

This article has been edited by the Royal Society of Chemistry, including the commissioning, peer review process and editorial aspects up to the point of acceptance.

# Heavy metal contamination and ecological risk assessment of the agricultural soil in Shanxi Province, China

Hongxue Qi, Bingqing Zhao, Lihong Li, Xiuling Chen, Jing An and Xiuping Liu

College of Chemistry and Chemical Engineering, Jinzhong University, Jinzhong, Shanxi 030619, People's Republic of China

(iD) LL, 0000-0002-4985-0570

To assess contamination levels and ecological risks of heavy metals in agricultural soil from Shanxi Province of China, a total of 33 samples in the surface soil were collected from 11 cities in Shanxi. The soil samples were digested by a mixed acid of nitric acid and hydrofluoric acid on a microwave digestion system, then the levels of eight heavy metals were analysed using an inductively coupled plasma mass spectrometer. The pollution levels of soil heavy metals were evaluated using a geo-accumulation index and their ecological risks were assessed using risk index calculated by Hakanson's method. As a result, the average concentrations of the heavy metals As, Cd, Cr, Cu, Hg, Ni, Pb and Zn were $12.9 \pm 4.8$, $0.35 \pm 0.23$, $43 \pm 14$, $27 \pm 19$, $0.25 \pm 0.14$, $21.7 \pm 5.7$, $17 \pm 13$ and $89 \pm 53$ mg kg$^{-1}$, respectively. By comparison to the *Chinese soil environmental quality* (GB15618-2018), only 9% of Cd samples and 3% of Cu samples exceeded their corresponding screening criteria. Subsequently, the results of geo-accumulation indices suggested that Shanxi's soil suffered from moderate to heavy contamination posed by Cd and Hg, and risk indices exhibited a similar trend that Cd and Hg were the main contributors for considerable to very high ecological risk. Finally, the analysis of variance indicated that the mean levels of Cd significantly occurred at Yuncheng areas among the 11 cities ($n = 3$, $p < 0.05$), but Hg concentrations did not have significantly statistical differences. This study demonstrated that metals Cd and Hg had higher levels and ecological risks for agricultural soil in Shanxi, especially, Yuncheng City suffered from heavy Cd contamination. The findings of the present study will provide basic information on management and control of the agricultural soil contamination in Shanxi Province, China.

# 1. Introduction

Soil is one of the most important sinks for the various pollutants, such as persistent organic pollutants [1], heavy metals [2] and emerging pollutants [3], and their accumulation could result in adverse effects on soil quality and productivity. Consequently, they could endanger various biological and food safety, and potentially threaten human health through the food chain.

Heavy metals are natural elements of the Earth's crust, and their levels in the soil are relatively low and basal level in reality. However, with anthropogenic input, heavy metal contaminations, e.g. arsenic (As), cadmium (Cd), chromium (Cr), copper (Cu), mercury (Hg), nickel (Ni), lead (Pb) and zinc (Zn), have become a severe problem in many parts of the world [2]. They are ubiquitous contaminants deposited in the environmental medium and accumulated in water bodies, soil, plants and even organisms. In addition, some metals (e.g. As, Cd, Cr, Ni, Hg and Pb) have been listed as a carcinogenic agency [4].

Shanxi Province in China covers a large area of agricultural land, and it is also well known for its coal resources. For instance, Shanxi's coal production accounted for 26.6% (25.2–29.0%) of China's coal production during 2010–2017 [5,6], and coal consumption also accounted for 11.4% (10.8–11.9%) synchronously [7,8] (electronic supplementary material, table S1). Especially, coal mining is considered as one of the most significant sources of heavy metal contamination in surrounding areas [9,10]. A similar pattern was found that heavy metal pollution resulted from metal As and its carcinogenic risk of human in Shanxi's mines [11]. Therefore, whether or not agricultural soil is safe has become a major concern. So far, some studies have reported that high levels of metals from soil were detected at Xiangfen County [12], Xinzhou area [13] and Fenhe Basin [14] in the Shanxi Province of China, indicating light pollution of soil contaminated metals in part areas. However, a full investigation for soil heavy metal pollution is relatively lacking in Shanxi of China.

The objectives of this study were fourfold: (i) determine the concentrations of heavy metals (including As, Cd, Cr, Cu, Hg, Ni, Pb and Zn) in the agricultural soil from the Shanxi Province, China; (ii) evaluate the status of contamination by using a geo-accumulation index ($I_{geo}$); (iii) assess the potential ecological risk; and (iv) distinguish the statistic differences of high-risk metals among the 11 cities in Shanxi.

# 2. Materials and methods

## 2.1. Study area and metal analysis

Sampling sites and soil collection were the same as assessing human cancer risk posed by polycyclic aromatic hydrocarbons [15]. As shown in the electronic supplementary material, figure S1, a total of 33 samples of agricultural soil were averagely collected from 11 cities in the Shanxi Province of China in 2018. The pH of each sample was detected by using an acidometer in a suspension of the soil into distilled water (1 : 2.5) [16]. The soil samples were digested by a mixed acid of nitric acid and hydrofluoric acid on a microwave digestion system according to the USEPA method 3052 [17], and then the concentrations of eight heavy metals were determined using an inductively coupled plasma mass spectrometer (ICP-MS; Agilent 7700). Briefly, the octopole collision reaction system for ICP-MS analysis was applied to eliminate mass spectral interference, the 0.18% L-cysteine was added into the sample solution to eliminate the memory effect of mercury [18], and all samples were spiked with an internal standard solution to monitor and correct for instrumental drift and matrix effects [19]. In addition, reagent blanks and soil standard reference material of China (GBW07408, GSS-8) were used for quality control and quality assurance. Detailed information on the metal analysis is shown in the electronic supplementary material.

## 2.2. Geo-accumulation index ($I_{geo}$)

The $I_{geo}$ has been used to evaluate the pollution level in soils since the late 1960s [20], it can be calculated using the following equation:

$$I_{geo} = \log_2\left(\frac{C_n}{1.5B_n}\right),$$

(2.1)

where $C_n$ is the measured concentration of every heavy metal in the soil (mg kg$^{-1}$), and $B_n$ is the geochemical background concentration of soil metal elements in Shanxi of China (mg kg$^{-1}$), which is given in the electronic supplementary material, table S2 [21].

**Table 1.** Statistical description of concentrations (mg kg$^{-1}$, $n = 33$), the criteria of the Chinese soil environmental quality and the percentages above their corresponding criteria of heavy metals from agricultural soil in 2018 in Shanxi Province, China.

|  | As | Cd | Cr | Cu | Hg | Ni | Pb | Zn |
|---|---|---|---|---|---|---|---|---|
| minimum | 4.83 | 0.09 | 27.2 | 11.4 | 0.039 | 14.0 | 7.52 | 32.9 |
| median | 11.3 | 0.27 | 40.1 | 21.5 | 0.25 | 19.6 | 13.4 | 74.0 |
| maximum | 23.2 | 1.18 | 93.1 | 107 | 0.77 | 36.4 | 67.6 | 300 |
| mean | 12.9 | 0.35 | 43 | 27 | 0.25 | 21.7 | 17 | 89 |
| s.d.[a] | 4.8 | 0.23 | 14 | 19 | 0.14 | 5.7 | 13 | 53 |
| control criteria[b] | 100 | 4.0 | 1300 | — | 6.0 | — | 1000 | — |
| screening criteria[b] | 25 | 0.6 | 250 | 100 | 3.4 | 190 | 170 | 300 |
| background[c] | 9.10 | 0.102 | 55.3 | 22.9 | 0.023 | 29.9 | 14.7 | 63.5 |
| (>control criteria) % | 0 | 0 | 0 | 0 | 0 | 0 | 0 | 0 |
| (>screening criteria) % | 0 | 9 | 0 | 3 | 0 | 0 | 0 | 0 |
| (>background) % | 79 | 97 | 15 | 45 | 100 | 15 | 36 | 67 |

[a]s.d.: standard deviation.
[b]Chinese soil environmental quality: risk control standard for soil contamination of agricultural land (GB 15618-2018, pH > 7.5) [23].
[c]Background: soil elements background values in Shanxi, China [21].

## 2.3. Ecological risk assessment

Hakanson's method was used to assess the potential ecological risks posed by the contamination of heavy metal in the soil [22]. Then the risk index (RI) for eight heavy metals was defined as

$$\mathrm{RI} = \sum E(i) = \sum T_i \times \frac{C_i}{C_{0i}}, \qquad (2.2)$$

where $E(i)$ is the potential ecological risk for an individual metal, $T_i$ is the toxic response factor for a given metal (As = 10, Cd = 30, Cr = 2, Cu = Ni = Pb = 5, Hg = 40, Zn = 1), $C_i$ is the concentration of a given $i$th metal in soil samples (mg kg$^{-1}$) and $C_{0i}$ is its corresponding background concentration (mg kg$^{-1}$).

## 2.4. Statistical analysis

Statistical differences of Cd and Hg concentrations among the 11 cities were analysed by one-way analysis of variance using SPSS 16.0 software. The multiple comparison tests were determined by a Tukey test, and significant differences were set at $p < 0.05$.

# 3. Results and discussion

## 3.1. Levels of heavy metals in soils

The levels of eight heavy metals from Shanxi's soils are presented in the electronic supplementary material, table S3 and their statistical description is presented in table 1. The average concentrations of the heavy metals As, Cd, Cr, Cu, Hg, Ni, Pb and Zn were 12.9 ± 4.8, 0.35 ± 0.23, 43 ± 14, 27 ± 19, 0.25 ± 0.14, 21.7 ± 5.7, 17 ± 13 and 89 ± 53 mg kg$^{-1}$, respectively. Furthermore, means of Cr and Ni were lower than their corresponding background values in Shanxi Province (table 1) [21], while remaining metals showed mean values greater than their background values. Especially, their samples of 79%, 97% and 100% exceeded the background values for As, Cd and Hg respectively. The concentrations of all the metals did not exceed the control criteria of *the Chinese soil environmental quality: risk control standard for soil contamination of agricultural land (GB 15 618-2018)* [23], but 9% of Cd samples (CZ2, YC1 and YC3) and 3% of Cu samples (YC3) exceeded their screening criteria (table 1).

To facilitate comparative analysis, mean concentrations of heavy metals in agricultural soils were obtained from other areas including worldwide, China and Shanxi Province of China (table 2). Meanwhile, the soil criteria were presented from Canada [29] and China [23].

**Table 2.** Mean concentrations (mg kg$^{-1}$) for heavy metals in agricultural soil compared to other countries and regions.

| country | area | samples | As | Cd | Cr | Cu | Hg | Ni | Pb | Zn | reference |
|---|---|---|---|---|---|---|---|---|---|---|---|
| USA | Baltimore | 22 | 3.7 | 0.32 | 18 | 18 | —[a] | 10 | 148 | 135 | [24] |
| India | Cavery Basin | 75 | — | 0.82 | 2.19 | 1.20 | — | 4.34 | 0.95 | 28.2 | [25] |
| Korea | Chungyang | 200 | 13.8 | 0.27 | — | 36.2 | — | 36.2 | 18.8 | 78.9 | [26] |
| Greece | Argolida | 132 | 6.95 | 0.54 | 83.1 | 74.7 | — | 146 | 19.7 | 74.9 | [27] |
| Spain | Duero Basin | 721 | — | 0.16 | 20.5 | 1.0 | 42.1 | 15.1 | 14.1 | 42.4 | [28] |
| Canada | guidelines | | 12 | 1.4 | 64 | 63 | — | 50 | 70 | 200 | [29] |
| China | screening criteria | | 25 | 0.6 | 250 | 100 | 3.4 | 190 | 170 | 300 | [23] |
| | | reveiw | 10.7 | 0.24 | 62.2 | 28.3 | 0.13 | 28.2 | 32.1 | 83.3 | [30] |
| | | 38 393 | 12.1 | 0.23 | 68.5 | 27.1 | 0.087 | 29.6 | 31.2 | 79.0 | [31] |
| | GZL, Jilin | 166 | — | 0.11 | 53.0 | 19.6 | — | 27.2 | 28.3 | 57.8 | [32] |
| | Chongqing City | 1,664 | 6.30 | 0.34 | 75.9 | 27.1 | 0.08 | 35.6 | 28.1 | 88.5 | [33] |
| | Hunan | 62–122 | 21.1 | 0.85 | 75.0 | 38.9 | 0.25 | 26.8 | 56.1 | 147 | [34] |
| | Changshu, Jiangsu | 105 | 7.46 | 0.11 | 86.4 | 31.6 | — | 34.9 | 31.4 | 61.1 | [35] |
| | Southern Fujian | 456 | 7.97 | — | 22.8 | 19.8 | — | 10.5 | 41.1 | 82.9 | [36] |
| | Hexi Corridor | 124 | — | — | 97.5 | 35.2 | — | 47.4 | 5.54 | 75.3 | [37] |
| | Puling, Guangdong | 413 | 7.89 | 0.06 | 22.5 | 11.1 | 0.08 | 12.0 | 42.4 | 56.5 | [38] |
| | Sihui, PRD[b] | 68 | 12.0 | — | — | 16.6 | 0.13 | 14.7 | 31.2 | — | [39] |
| | Shunde, PRD | 29 | 21.6 | — | — | 44.2 | 0.45 | 35.4 | 48.3 | — | [39] |
| | PRD | 2,241 | 13.0 | 0.27 | 51.8 | — | 0.26 | — | 47.3 | — | [40] |
| Shanxi, | Xiangfen | 128 | 14.0 | 0.20 | 71.4 | 30.1 | 0.13 | 32.4 | 23.9 | 82.6 | [41] |
| China | Xinzhou | 247 | 6.5 | 0.11 | 59 | 22 | — | 27 | 20 | 72 | [13] |
| | Fenhe Basin | 50 | 15.8 | 0.19 | 51.4 | 21.5 | 0.05 | 27.5 | 22.1 | 72.3 | [14] |
| | whole province | 33 | 12.9 | 0.35 | 43.4 | 27.3 | 0.25 | 21.7 | 17.0 | 89.2 | this study |

[a]—: not reported.
[b]PRD: Pearl River Delta.

**Table 3.** Classes of geo-accumulation index ($I_{geo}$) and its percentage distribution for heavy metals in the surface soil in Shanxi Province, China (%, $n = 33$).

| class | soil quality | range | As | Cd | Cr | Cu | Hg | Ni | Pb | Zn |
|---|---|---|---|---|---|---|---|---|---|---|
| 0 | uncontaminated | $I_{geo} < 0$ | 67 | 6 | 97 | 85 | 0 | 100 | 85 | 73 |
| 1 | uncontaminated to moderately contaminated | $0 \leq I_{geo} < 1$ | 33 | 52 | 3 | 9 | 6 | 0 | 9 | 21 |
| 2 | moderately contaminated | $1 \leq I_{geo} < 2$ | 0 | 33 | 0 | 6 | 12 | 0 | 6 | 6 |
| 3 | moderately to heavily contaminated | $2 \leq I_{geo} < 3$ | 0 | 9 | 0 | 0 | 48 | 0 | 0 | 0 |
| 4 | heavily contaminated | $3 \leq I_{geo} < 4$ | 0 | 0 | 0 | 0 | 30 | 0 | 0 | 0 |
| 5 | heavily to extremely contaminated | $4 \leq I_{geo} < 5$ | 0 | 0 | 0 | 0 | 3 | 0 | 0 | 0 |
| 6 | extremely contaminated | $5 \leq I_{geo}$ | 0 | 0 | 0 | 0 | 0 | 0 | 0 | 0 |

Worldwide, agricultural soil was hardly polluted by heavy metals in USA [24], India [25] and Korea [26]. However, heavy metal contamination in Greece was relatively higher than other countries in agricultural soil [27], whereas Hg contamination in Spain was relatively serious [28].

In China, although the comparison of the average value indicated the variation in heavy metal concentrations among the different regions, the levels did not exceed the screening criteria of the *Chinese soil environmental quality* (GB 15618-2018) [23] except for Cd (0.6 versus 0.85 mg kg$^{-1}$) in Hunan Province [34]. Furthermore, the mean concentrations of all metals in soil from Hunan Province were higher than the mean values found in Shanxi of China.

In the Shanxi Province of China, the concentrations of soil metal detected in this study were noticeably lower than those in Xiangfen County [12] and Fenhe Basin [14], while metal contents [13] in Xinzhou City were accordant with the results found in the present study.

Overall, mean contents of soil heavy metal, compared with the previous studies, were generally in the lower levels in Shanxi of China, but Shanxi severely suffered from Cd contamination in the agricultural areas.

## 3.2. Pollution assessment of soil metals using geo-accumulation index

The $I_{geo}$ for heavy metals in soils were calculated in the electronic supplementary material, table S2 and their percentage distribution is presented in table 3. According to Muller's approach [20], the $I_{geo}$ consists of seven classes. Conditions of Cd and Hg were apparently severe, whereas the remaining six metals were uncontaminated to moderately contaminated levels ($I_{geo} < 1$).

In detail, 52% of samples of Cd accumulation regarded as uncontaminated to moderately contaminated ($0 \leq I_{geo} < 1$), 33% as moderately contaminated ($1 \leq I_{geo} < 2$) and 9% as moderately to heavily contaminated ($2 \leq I_{geo} < 3$). By contrast, the situation of Hg was more serious than Cd contamination. Only 6% of samples of Hg accumulation presented as uncontaminated to moderately contaminated ($0 \leq I_{geo} < 1$), 12% as moderately contaminated ($1 \leq I_{geo} < 2$), 48% as moderately to heavily contaminated ($2 \leq I_{geo} < 3$), 30% as heavily contaminated ($3 \leq I_{geo} < 4$) and 3% as heavily to extremely contaminated ($4 \leq I_{geo} < 5$). The above results suggested that most of the sites were in the moderate to heavy contamination posed by Cd and Hg in Shanxi's soils, which was consistent with the previous report that Cd and Hg have been regarded as the priority control metals because of their higher levels [31]. In the future, the metal contamination posed by Cd and Hg should be to managed and controlled strictly in the study area.

## 3.3. Ecological risk assessment of soil metals

The potential ecological risk $E(i)$ values of eight heavy metals in soil are shown in the electronic supplementary material, table S4, and their corresponding percentages are presented in table 4. According to Hakanson's approach [22], the RI values of 6% of all samples are regarded as moderate risk ($150 \leq RI < 300$), while 58% as considerable risk ($300 \leq RI < 600$), 33% as very high risk ($600 \leq RI < 1200$) and 3% as dangerous ($\geq 1200$) potential ecological risk.

**Table 4.** Grades of potential ecological risk and their corresponding percentages for heavy metals in agricultural soils in 2018 in Shanxi Province of China (%, n = 33).

| E(i) [a] | As | Cd | Cr | Cu | Hg | Ni | Pb | Zn | ecological risk | RI [b] | % |
|---|---|---|---|---|---|---|---|---|---|---|---|
| E(i) < 40 | 100 | 3 | 100 | 100 | 0 | 100 | 100 | 100 | low | RI < 150 | 0 |
| 40 ≤ E(i) < 80 | 0 | 48 | 0 | 0 | 3 | 0 | 0 | 0 | moderate | 150 ≤ RI < 300 | 6 |
| 80 ≤ E(i) < 160 | 0 | 39 | 0 | 0 | 6 | 0 | 0 | 0 | considerable | 300 ≤ RI < 600 | 58 |
| 160 ≤ E(i) < 320 | 0 | 6 | 0 | 0 | 21 | 0 | 0 | 0 | very high | 600 ≤ RI < 1200 | 33 |
| 320 ≤ E(i) | 0 | 3 | 0 | 0 | 70 | 0 | 0 | 0 | dangerous | 1200 ≤ RI | 3 |

[a]E(i): the scope of potential ecological risk factor for the given metal.

[b]RI: the scope of integrated potential ecological risk index for all the metal in a region.

**Table 5.** The analysis of variance for Cd and Hg concentrations (mg kg$^{-1}$) among the 11 cities in Shanxi Province, China.

| | | cities | Cd | | Hg | |
|---|---|---|---|---|---|---|
| | | | mean ± s.d.[a] | $p < 0.05$ | mean ± s.d. | $p > 0.05$ |
| 1 | DT | Datong | 0.30 ± 0.12 | AB[b] | 0.40 ± 0.34 | A |
| 2 | XZ | Xinzhou | 0.22 ± 0.04 | B | 0.28 ± 0.11 | A |
| 3 | SZ | Shuozhou | 0.26 ± 0.15 | B | 0.24 ± 0.02 | A |
| 4 | TY | Taiyuan | 0.22 ± 0.15 | B | 0.32 ± 0.12 | A |
| 5 | YQ | Yangquan | 0.24 ± 0.06 | B | 0.22 ± 0.17 | A |
| 6 | JZ | Jinzhong | 0.24 ± 0.02 | B | 0.26 ± 0.04 | A |
| 7 | LL | Lvliang | 0.25 ± 0.04 | B | 0.28 ± 0.05 | A |
| 8 | CZ | Changzhi | 0.53 ± 0.24 | AB | 0.17 ± 0.09 | A |
| 9 | LF | Linfen | 0.34 ± 0.06 | AB | 0.22 ± 0.18 | A |
| 10 | JC | Jincheng | 0.41 ± 0.09 | AB | 0.20 ± 0.12 | A |
| 11 | YC | Yuncheng | 0.81 ± 0.48 | A | 0.21 ± 0.16 | A |

[a]s.d.: standard deviation ($n = 3$).
[b]Statistical differences were determined by the analysis of variance and multiple comparison tests were performed by Tukey test, and the same letter on the same column indicated that they were not significantly different.

In terms of composition, the high ecological risks mainly resulted from the contribution of Cd and Hg. The maximum values of $E(i)$ of six metals except for Cd and Hg were all less than 26 (electronic supplementary material, table S4), which demonstrated that these metals posed a relatively low potential ecological risk. However, Cd and Hg had much higher $E(i)$ mean values (Cd: 99.5, Hg: 429, electronic supplementary material, table S4) than those other six heavy metals. For instance, 3% of all samples of $E$(Cd) regarded as low risk ($E(i)$ less than 40), while 48% as moderate risk ($40 \leq E(i) < 80$), 39% as considerable risk ($80 \leq E(i) < 160$), 6% as very high risk ($160 \leq E(i) < 320$) and 3% as dangerous ($E(i) > 320$). Analogously, 3% of all samples of $E$(Hg) presented as the moderate risk ($40 \leq E(i) < 80$), while 6% as considerable risk ($80 \leq E(i) < 160$), 21% as very high risk ($160 \leq E(i) < 320$) and even 70% as potential dangerous ecological risk ($E(i) \geq 320$).

To sum up, the accumulations of metals Cd and Hg in agricultural soil in Shanxi posed a potential high risk to the ecosystem, which was consistent with similar trends in previous studies. For example, metals Cd and Hg showed a higher ecological risk than the other metals from a typical county (Xiangfen) in Shanxi [41] and from the middle reaches of Fenhe River Basin in Shanxi [14]. Moreover, similar findings were also reported on China from the first national soil pollution survey during April 2005 to December 2013 [31] and a meta-analysis from 2005 to 2017 [30]. Overall, most of the sampling sites in Shanxi of China suffered from the considerable, even very high ecological risk posed by heavy metals Cd and Hg. These two metals should be listed as the priority control contaminants in the future.

## 3.4. Spatial distribution

Given Cd and Hg were identified as the main contributors for the above analysis, the analysis of variance was performed by average values to distinguish the statistical differences of Cd and Hg concentrations among the 11 cities.

### 3.4.1. Spatial distribution of Cd concentrations

As shown in table 5, the highest Cd content ($0.81 \pm 0.48$ mg kg$^{-1}$) was found in Yuncheng City ($n = 3$, $p < 0.05$), where the average concentration was significantly higher than those of Shuozhou, Xinzhou, Taiyuan, Yangquan, Jinzhong and Lvliang, while those of the remainder (Datong, Changzhi, Linfen and Jincheng) were at the median levels. Apparently, the maximum Cd concentration (Yuncheng area) was up to eightfold greater than the background values in Shanxi Province of China. Cd deposition in agricultural soil subsequently bio-accumulates in food crops. For instance, high-level Cd in the soil can accumulate in rice grain [42]. Long-term intake of Cd can accumulate in nearly every organ

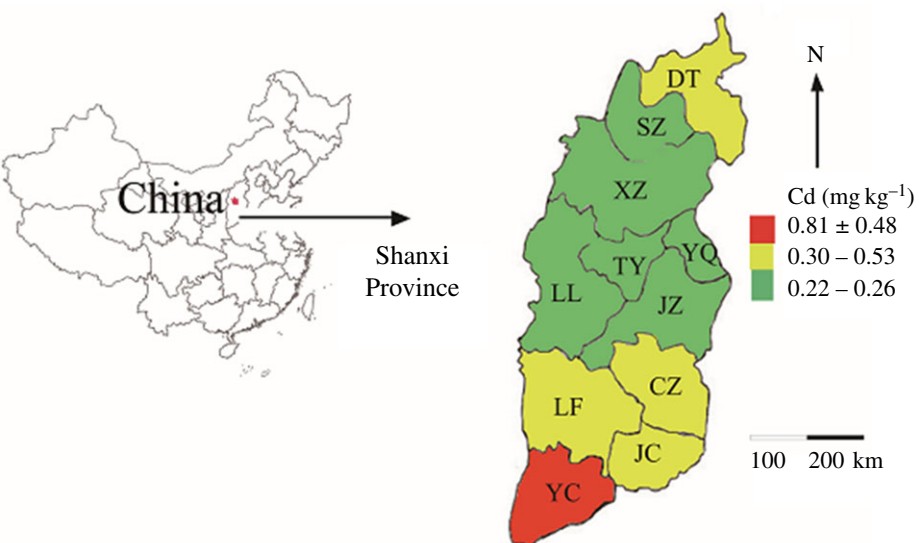

**Figure 1.** Spatial distribution of heavy metal Cd in agricultural soils in Shanxi Province, China (DT: Datong, SZ: Shuozhou, XZ: Xinzhou, TY: Taiyuan, YQ: Yangquan, LL: Lvliang, JZ: Jinzhong, LF: Linfen, CZ: Changzhi, YC: Yuncheng, JC: Jincheng).

and tissue in the human body [43] and finally result in serious toxic and side effects, such as renal dysfunction [44].

Compared to other provinces in China, Cd contents in Yuncheng were at a relatively low level [45]. In fact, Cd concentrations in soil accumulated gradually from 1981 to 2016 in China [45], which were mainly from smelting, mining, fertilizer, pesticide, sewage irrigation, waste disposal and vehicle exhaust [46]. Among them, the pollution levels of heavy metals at the point source (e.g. coal mine) was obviously higher than those in the surrounding areas [11]. In addition, the levels of heavy metals in urban areas and cities were much higher than those in agricultural regions [47]. Therefore, point source pollution cannot be ignored for controlling and remediating soil contamination in the future.

### 3.4.2. Spatial distribution of Hg concentrations

Surprisingly, no significantly statistical difference in Hg concentrations was found among the 11 cities ($n = 3$, $p > 0.05$) (table 5). To a certain extent, this negative result suggested that the new point source was not the main contributor to Hg contamination in agricultural soil from the Shanxi Province of China. Because if there had been a new point source input before, the Hg levels in this region would have shown a statistical difference from other regions. Therefore, Hg accumulations were no new point source input but had higher levels in the whole area in Shanxi Province, the reasonable explanation is that the primary source of Hg probably resulted from atmospheric deposition. Namely, Hg emissions from the point sources (e.g. mines or industrial areas [48]) emit into the air, then disperses the long distance, and finally deposits in other areas [49]. Because atmospheric deposition was more inclined to the large-scale and relative homogenizing contamination, other pathways, e.g. agrochemicals, fertilizers, livestock manure and sewage irrigation, were more inclined to point source pollution. For instance, a previous study also reported that atmospheric deposition was a main pathway to the various pollution sources, it accounted for approximately 61% of the total annual input of Hg in agricultural soil in China [50].

In addition, for the potential ecological risk, the $E$(Hg) was found to be greater than 320 in 70% of the samples in Shanxi Province; analogously, the $E$(Hg) was calculated to be greater than 320 in 36% of the samples in agricultural soils throughout China [51], suggesting that not only Shanxi Province but also the whole of China suffers from the high potential ecological risk posed by Hg accumulation.

Overall, Cd accumulations showed higher levels and ecological risk from Yuncheng areas among the 11 cities (figure 1), but the spatial distribution of Hg was not a significantly statistical difference. Until now, many technologies have emerged to remediate the metal contamination, such as phytoremediation, ecological and hydraulic remediation [52], and organic amendments [53]. However, the treatment of heavy metal pollution is still a thorny issue. Therefore, it is very urgent to develop a soil remediation technique with consideration to the type and degree of pollution, field characteristics, remediation targets, implementation schedule, cost effectiveness and public acceptability [54].

# 4. Conclusion

In the present study, the levels of eight heavy metals (As, Cd, Cr, Cu, Hg, Ni, Pb and Zn) were detected and their potential ecological risks were estimated in agricultural soil from Shanxi Province in China. Among them, the metals Cd and Hg had higher levels and ecological risk for agricultural soil in Shanxi. Among the 11 cities, the average concentration of Cd in Yuncheng City was significantly higher than those in the other sites, but the spatial distribution of Hg was not a significantly statistical difference. However, the levels of Hg in Shanxi did not exceed the soil screening criteria (3.4 mg kg$^{-1}$) in China (GB 15618-2018) [23], 100% samples of Hg exceeded the background values and the average value of Hg (0.25 ± 0.14 mg kg$^{-1}$) was about ten times higher than its background value (0.023 mg kg$^{-1}$). These results urge us to weaken the coal consumption and control point source pollution for the safety of the agricultural soil in Shanxi Province, China.

Data accessibility. Electronic supplementary material is available online at rs.figshare.com. Detailed methods for soil metal analyses are included in the electronic supplementary material. The concentrations of eight heavy metals, geo-accumulation index and the potential ecological risk are presented in the electronic supplementary material, tables S1–S4.

Authors' contribution. H.Q. participated in the design of the study and drafted the manuscript; B.Z. and X.L. carried out the statistical analyses and critically revised the manuscript; X.C. and J.A. collected soil samples and critically revised the manuscript; L.L. designed and coordinated the study and helped draft the manuscript. All authors gave final approval for publication and agree to be held accountable for the work performed therein.

Competing interests. The authors declare no competing interests.

Funding. This work was supported by the Natural Science Foundation of Shanxi Province, China (201901D111299 and 201901D111301), Scientific and Technological Innovation Programs of Higher Education Institutions in Shanxi, China (2020L0584), the Program for '1331 Project' Key Innovative Research Team of Shanxi Province, China (PY201817), the Program for '1331 Project' Key Innovative Research Team of Jinzhong University (jzxycxtd2017007), the Teaching Reform and Innovation Foundation for the Higher Education Institutions of Shanxi Province, China (J2019021), and the Program for '1331 Project' Maker Team of Jinzhong University (jzxycktd2019031).

Acknowledgement. We thank all the students from applied chemistry 1501 of Jinzhong University for assistance in soil collection.

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
