## [Reviewer comments · Royal Society Open Science]

Review History

RSOS-200538.R0 (Original submission)

Review form: Reviewer 1

Is the manuscript scientifically sound in its present form?

Yes

Are the interpretations and conclusions justified by the results?

Yes

Is the language acceptable?

Yes

Do you have any ethical concerns with this paper?

No

Have you any concerns about statistical analyses in this paper?

No

Recommendation?

Accept with minor revision (please list in comments)

Comments to the Author(s)

Proof reading of the manuscript is highly recommended for the International audience and availability of information mentioned in the text but not found in the direct document

Review form: Reviewer 2

Is the manuscript scientifically sound in its present form?

Yes

Are the interpretations and conclusions justified by the results?

Yes

Is the language acceptable?

Yes

Do you have any ethical concerns with this paper?

No

Have you any concerns about statistical analyses in this paper?

No

Recommendation?

Major revision is needed (please make suggestions in comments)

Comments to the Author(s)

Manuscript ID RSOS-200538

Manuscript title: Heavy metals contamination and ecological risk assessment of the agricultural soil in Shanxi Province, China

Authors: Hongxue Qi, Bingqing Zhao, Lihong Li*, Xiuling Chen, An Jing, Xiuping Liu

The manuscript presents the levels of eight heavy metals (As, Cd, Cr, Cu, Hg, Ni, Pb, and Zn) in agricultural soils from Shanxi Province of China. Elemental analysis was performed by inductively coupled plasma mass spectrometer (ICP-MS). Potential ecological risks were estimated.

It is needed to present clearer description of procedures that were applied on this study to make the manuscript more understandable.

- ICP-MS suffers from unwanted spectral and nonspectral, which might change with the elemental composition of the samples analysed. The elements that form the polyatomic interferences (including C, Cl, H, N and O) usually result from the sample matrix, argon plasma, nebulized water and dissolved/entrained air. Most of the interference can be reduced by using a collision-reaction interface (CRI) or dynamic reaction cells, while other would require a sector field ICP-MS. The authors should specify for As, Hg and Cr which systems were used to control or eliminate interference during the measurements.

- Memory effects are major challenges in Hg determination by ICP-MS. How was it checked?

- An internal standard (IS) is important for monitoring the change in the ICP-MS signal due to the plasma instability and blockage of the orifice of the sample cone, etc. Unfortunately, I found no information and discussion regarding the use of IS.

- Significant figures must be corrected along the text and tables. Concentration values and standard deviation must be written to not more than three and two significant figures, respectively.

Example: on page 3, line 20: 12.9 ± 4.83 mg/kg  12.9 ± 4.8 mg/kg; 89.2 ± 53.2 mg/kg  89 ± 53 mg/kg.

- On page 4, line 54: Xiangfen county

Decision letter (RSOS-200538.R0)

Dear Dr Li:

Title: Heavy metals contamination and ecological risk assessment of the agricultural soil in Shanxi Province, China
Manuscript ID: RSOS-200538

The editor assigned to your manuscript has now received comments from reviewers. We would like you to revise your paper in accordance with the referee and Subject Editor suggestions which can be found below (not including confidential reports to the Editor). Please note this decision does not guarantee eventual acceptance.

Please submit your revised paper before 05-Jul-2020. Please note that the revision deadline will expire at 00.00am on this date. If we do not hear from you within this time then it will be assumed that the paper has been withdrawn. In exceptional circumstances, extensions may be possible if agreed with the Editorial Office in advance. We do not allow multiple rounds of revision so we urge you to make every effort to fully address all of the comments at this stage. If deemed necessary by the Editors, your manuscript will be sent back to one or more of the original reviewers for assessment. If the original reviewers are not available we may invite new reviewers.

On behalf of the Subject Editor Professor Anthony Stace and the Associate Editor Dr Nadia Martinez Villegas.

RSC Associate Editor:

Comments to the Author:

This manuscript presents an interesting study on trace metal contamination of agricultural soil in Shanxi, China. Nevertheless, methods regarding ICP analyses must be improved in order to prove to be sound and to give reliable data to allow replication; specially for Hg analysis. Additionally, the discussion on the metal contaminants should consider the geology of the area to put into perspective places being compared to. Why is that basically all the study area is enriched with Hg? Finally, the supporting information must be provided. Authors need to do proof reading for their sentences in order to communicate effectively and correctly the science in the work.

RSC Subject Editor:

Comments to the Author:

(There are no comments.)

Reviewers' Comments to Author:

Reviewer: 1

Comments to the Author(s)

Proof reading of the manuscript is highly recommended for the International audience and availability of information mentioned in the text but not found in the direct document

Reviewer: 2

Comments to the Author(s)

Royal Society Open Science
Manuscript ID RSOS-200538

Manuscript title: Heavy metals contamination and ecological risk assessment of the agricultural soil in Shanxi Province, China

Authors: Hongxue Qi, Bingqing Zhao, Lihong Li*, Xiuling Chen, An Jing, Xiuping Liu

The manuscript presents the levels of eight heavy metals (As, Cd, Cr, Cu, Hg, Ni, Pb, and Zn) in agricultural soils from Shanxi Province of China. Elemental analysis was performed by

inductively coupled plasma mass spectrometer (ICP-MS). Potential ecological risks were estimated.

It is needed to present clearer description of procedures that were applied on this study to make the manuscript more understandable.

- ICP-MS suffers from unwanted spectral and nonspectral, which might change with the elemental composition of the samples analysed. The elements that form the polyatomic interferences (including C, Cl, H, N and O) usually result from the sample matrix, argon plasma, nebulized water and dissolved/entrained air. Most of the interference can be reduced by using a collision-reaction interface (CRI) or dynamic reaction cells, while other would require a sector field ICP-MS. The authors should specify for As, Hg and Cr which systems were used to control or eliminate interference during the measurements.

- Memory effects are major challenges in Hg determination by ICP-MS. How was it checked?

- An internal standard (IS) is important for monitoring the change in the ICP-MS signal due to the plasma instability and blockage of the orifice of the sample cone, etc. Unfortunately, I found no information and discussion regarding the use of IS.

- Significant figures must be corrected along the text and tables. Concentration values and standard deviation must be written to not more than three and two significant figures, respectively.

Example: on page 3, line 20: 12.9 ± 4.83 mg/kg  12.9 ± 4.8 mg/kg; 89.2 ± 53.2 mg/kg  89 ± 53 mg/kg.

- On page 4, line 54: Xiangfen county

Author's Response to Decision Letter for (RSOS-200538.R0)

See Appendix A.

RSOS-200538.R1 (Revision)

Review form: Reviewer 2

Is the manuscript scientifically sound in its present form?

No

Are the interpretations and conclusions justified by the results?

Yes

Is the language acceptable?

No

Do you have any ethical concerns with this paper?

Yes

Have you any concerns about statistical analyses in this paper?

Yes

Recommendation?

Accept with minor revision (please list in comments)

Comments to the Author(s)

Compared to the first draft, the authors have considerably improved the manuscript. Most of the comments have been taken into account. It is my opinion that the manuscript is worthy of publication in Royal Society Open Science after addressing the following issues:

- Regarding a response to the comments, I would like to highlight that the USEPA method 3052 has several additional alternative acid and reagent combinations including hydrochloric acid and hydrogen peroxide and only the reagent mixture $\text{HNO}_3/\text{H}_2\text{O}_2$ (or HNO_3/HF) avoid the addition of Cl (HCl).
- In the "Supporting Information", the authors report the following digestion method: "The digestion was carried out on a microwave digestion system (WX-6000, PreeKem Scientific Instruments Co., Ltd.) with a digestion solution of 7 ml of nitric acid and 3 ml of hydrofluoric acid"; while on page 3, lines 70-71 of the revised manuscript "The soil samples were digested by a mixed acid of nitric acid and hydrochloric acid on a microwave digestion system according to the USEPA method 3052". Which reagent mixture for microwave assisted digestion has been used? HNO_3/HCl or HNO_3/HF ?
- Supporting Information: Table S1: the Hg data is not present; on page S3 line 47 "The detect limitation for eight metals were 0.011 – 0.038 mg/kg dw" should be "The detection limits for eight metals were in the range of 0.011 – 0.038 mg/kg dw"; line 39: 97.6% – 106.2% should be 97.6 – 106%; 78% – 129% should be 78 – 129%. etc.
- Abstract: Significant figures must still be corrected along the text, tables and "Supporting Information":
 43.4 ± 14 mg/kg should be 43 ± 14 mg/kg, 27.3 ± 19 mg/kg should be 27 ± 19 mg/kg; 17.0 ± 13 mg/kg should be 17 ± 13 mg/kg, and 89.2 ± 53 mg/kg should be 89 ± 53 ; lines 48-50: 26.63% (25.21%-29.02%) should be 26.6% (25.2-29.0%) etc....
- On page 3, line 50: Table S2: Number tables consecutively in accordance with their appearance in the text (Table S1, S2 etc).

Decision letter (RSOS-200538.R1)

Dear Dr Li:

Title: Heavy metals contamination and ecological risk assessment of the agricultural soil in Shanxi Province, China
 Manuscript ID: RSOS-200538.R1

Thank you for submitting the above manuscript to Royal Society Open Science. On behalf of the Editors and the Royal Society of Chemistry, I am pleased to inform you that your manuscript will be accepted for publication in Royal Society Open Science subject to minor revision in accordance with the referee suggestions. Please find the reviewers' comments at the end of this email.

The reviewers and handling editors have recommended publication, but also suggest some minor revisions to your manuscript. Therefore, I invite you to respond to the comments and revise your manuscript.

Because the schedule for publication is very tight, it is a condition of publication that you submit the revised version of your manuscript before 19-Aug-2020. Please note that the revision deadline will expire at 00.00am on this date. If you do not think you will be able to meet this date please let me know immediately.

Kind regards,
Dr Laura Smith
Publishing Editor, Journals

Royal Society of Chemistry
Thomas Graham House
Science Park, Milton Road

Cambridge, CB4 0WF
Royal Society Open Science - Chemistry Editorial Office

On behalf of the Subject Editor Professor Anthony Stace and the Associate Editor Dr Nadia Martinez Villegas.

RSC Associate Editor:

Comments to the Author:

Thank you very much for the effort to improve this manuscript. A few more issues need to be addressed before your manuscript is ready for publicación. Please address the comments for Reviewer 2 and the following: The format of the abstract must meet the standard requirements of objectives, materials and methods, results and conclusions. In its current version, the materials and methods section are incomplete. Additionally, a professional proofreading service is highly recommended as the language needs a thorough revision.

RSC Subject Editor:

Comments to the Author:

(There are no comments.)

Reviewer comments to Author:

Reviewer: 2

Comments to the Author(s)

Compared to the first draft, the authors have considerably improved the manuscript. Most of the comments have been taken into account. It is my opinion that the manuscript is worthy of publication in Royal Society Open Science after addressing the following issues:

- Regarding a response to the comments, I would like to highlight that the USEPA method 3052 has several additional alternative acid and reagent combinations including hydrochloric acid and hydrogen peroxide and only the reagent mixture $\text{HNO}_3/\text{H}_2\text{O}_2$ (or HNO_3/HF) avoid the addition of Cl (HCl).

- In the "Supporting Information", the authors report the following digestion method: "The digestion was carried out on a microwave digestion system (WX-6000, PreeKem Scientific Instruments Co., Ltd.) with a digestion solution of 7 ml of nitric acid and 3 ml of hydrofluoric acid"; while on page 3, lines 70-71 of the revised manuscript "The soil samples were digested by a mixed acid of nitric acid and hydrochloric acid on a microwave digestion system according to the USEPA method 3052". Which reagent mixture for microwave assisted digestion has been used? HNO_3/HCl or HNO_3/HF ?

- Supporting Information: Table S1: the Hg data is not present; on page S3 line 47 "The detection limitation for eight metals were 0.011 - 0.038 mg/kg dw" should be "The detection limits for eight metals were in the range of 0.011 - 0.038 mg/kg dw"; line 39: 97.6% - 106.2% should be 97.6 - 106%; 78% - 129% should be 78 - 129%. etc.

- Abstract: Significant figures must still be corrected along the text, tables and "Supporting Information":

43.4 ± 14 mg/kg should be 43 ± 14 mg/kg, 27.3 ± 19 mg/kg should be 27 ± 19 mg/kg; 17.0 ± 13 mg/kg should be 17 ± 13 mg/kg, and 89.2 ± 53 mg/kg should be 89 ± 53 ; lines 48-50: 26.63% (25.21%-29.02%) should be 26.6% (25.2-29.0%) etc....

- On page 3, line 50: Table S2: Number tables consecutively in accordance with their appearance in the text (Table S1, S2 etc).

Author's Response to Decision Letter for (RSOS-200538.R1)

See Appendix B.

Decision letter (RSOS-200538.R2)

Dear Dr Li:

Title: Heavy metals contamination and ecological risk assessment of the agricultural soil in Shanxi Province, China
Manuscript ID: RSOS-200538.R2

It is a pleasure to accept your manuscript in its current form for publication in Royal Society Open Science. The chemistry content of Royal Society Open Science is published in collaboration with the Royal Society of Chemistry.

Yours sincerely,
Dr Ellis Wilde
Publishing Editor, Journals

On behalf of the Subject Editor Professor Anthony Stace and the Associate Editor Dr Nadia Martinez Villegas.

RSC Associate Editor

Comments to the Author:

Thank you very much for the revised version of the manuscript. Changes were made up to the satisfaction of the reviewers.

Reviewer(s)' Comments to Author:

Appendix A

June 24, 2020

Dear Dr. Laura Smith,

Thank you very much for your favorable consideration and for the insightful comments of Associate Editor and reviewers concerning our manuscript entitled “Heavy metals contamination and ecological risk assessment of the agricultural soil in Shanxi Province, China” (RSOS-200538). Those comments are very helpful for improving our manuscript. Here below is our description on the revisions according to the Associate Editor and reviewers with all corrections being **highlighted** in the text. We list all the changes in the response to the reviewer as follows. In addition, the **tables were moved** from the end of the article to the nearest place to the text description. We hope you find the revised manuscript suitable for publication in *Royal Society Open Science*.

Once again, thank you very much for your time and consideration.

Sincerely,

Lihong Li

lilh@jzxy.edu.cn

Responses to the comments

RSC Associate Editor:

This manuscript presents an interesting study on trace metal contamination of agricultural soil in Shanxi, China. Nevertheless, methods regarding ICP analyses must be improved in order to prove to be sound and to give reliable data to allow replication; specially for Hg analysis.

Response: We thank the RSC Associate Editor for the constructive suggestion. This is an

important reminder of us to specify the information on the ICP-MS analyses. Now, this information was supplemented as follows and inserted into the text accordingly (Lines 70-78).

And three references have been added.

“The soil samples were digested by a mixed acid of nitric acid and hydrochloric acid on a microwave digestion system according to the USEPA method 3052 [1], and the concentrations of eight heavy metals were determined using an inductively coupled plasma mass spectrometer (ICP-MS, Agilent 7700). Briefly, the octopole collision-reaction system for ICP-MS analysis was applied to eliminate mass spectral interference, the 0.18% L-cysteine was added into the sample solution to eliminate the memory effect of mercury[2], and all samples were spiked with internal standard solution to monitor and correct for instrumental drift and matrix effects[3]. In addition, reagent blanks and soil standard reference material of China (GBW07408, GSS-8) were used for quality control and quality assurance.”

Additionally, the discussion on the metal contaminants should consider the geology of the area to put into perspective places being compared to. Why is that basically all the study area is enriched with Hg?

Response: We really appreciate the reviewer's comments. Background value of Hg is 0.023 mg/kg in Shanxi province (Table 1), while it is 0.06 mg/kg in China [7]. The background values are always considered to calculate the geo-accumulation index (I_{geo}) and the potential ecological risk.

Despite the levels of Hg in Shanxi did not exceed the screening criteria (3.4 mg/kg) of *the Chinese soil environmental quality: risk control standard for soil contamination of agricultural land (GB 15618-2018)* [23], 100% samples exceeded the background values for Hg and the average value of Hg (0.25 ± 0.14 mg/kg) was about ten times higher than its background value

(0.023 mg/kg). Moreover, no significantly statistical difference in Hg concentrations was found among the 11 cities ($n = 3$, $p > 0.05$) (Table 5).

All the study area is enriched with Hg, the reasonable explanation was that the primary source of Hg probably resulted from atmospheric deposition. And the corresponding description were presented as follows and at the text (226-236).

“Therefore, Hg accumulations were no new point source input but had higher levels in the whole areas in Shanxi Province, the reasonable explanation was that the primary source of Hg probably resulted from atmospheric deposition. Namely, Hg emissions from point sources (e.g. mines or industrial area [4]) to air, then disperses the long distance, finally deposits the other areas [5]. Because atmospheric deposition was more inclined to large-scale and nearly homogenizing contamination, while other pathways, e.g. agrochemicals, fertilizers, livestock manure and sewage irrigation, were more inclined to point source pollution. For instance, the previous study reported that atmospheric deposition was a main pathway to the various pollution sources, it accounted for approximately 61% of the total annual input of Hg in agricultural soil in China [6].”

Finally, the supporting information must be provided. Authors need to do proof reading for their sentences in order to communicate effectively and correctly the science in the work.

Response: Now the supporting information has been provided as an independent file. And the manuscript has been thoroughly revised according to the suggestions. Detail information is listed as follow:

An author name “An Jing” has been changed to “Jing An” (Line 3).

“the levels of” has been inserted to clarify the sentence meaning (Line 18).

“concentration wasn’t” has been changed to “concentrations weren’t” (Line 28).

“management and control” have been inserted to clarify the sentence meaning (Line 31).

“with” has been changed to “in” (Line 43).

“metal” has been changed to “metals” (Line 45).

“chemical” has been changed to “metal” (Line 65).

“these high risks” has been changed to “high potential ecological risk” (Line 172).

“the other six heavy metal” has been changed to “others six heavy metals” (Line 176).

“poses” has been changed to “posed” (Line 182).

“Cd contents (0.81 ± 0.48 mg/kg) were” has been changed to “Cd content (0.81 ± 0.48 mg/kg) was” (Line 196).

“that” has been changed to “those” (Line 209).

“For” has been changed to “In addition, for” (Line 237).

“average concentration for Cd in Yuncheng city was significantly higher than those in other sites” has been changed to “the average concentration of Cd in Yuncheng city was significantly higher than those in the other sites” (Line 259-260).

A funding of “Scientific and Technological Innovation Programs of Higher Education Institutions in Shanxi, China (2020L0584)” has been added (Lines 279-280).

Reviewer: 1

Proof reading of the manuscript is highly recommended for the International audience and availability of information mentioned in the text but not found in the direct document.

Response: We really appreciate the reviewer's suggestions. The manuscript has been extensively revised according to the suggestions. And the supporting information has been provided as an

independent file.

Reviewer: 2

The manuscript presents the levels of eight heavy metals (As, Cd, Cr, Cu, Hg, Ni, Pb, and Zn) in agricultural soils from Shanxi Province of China. Elemental analysis was performed by inductively coupled plasma mass spectrometer (ICP-MS). Potential ecological risks were estimated. It is needed to present clearer description of procedures that were applied on this study to make the manuscript more understandable.

- ICP-MS suffers from unwanted spectral and nonspectral, which might change with the elemental composition of the samples analysed. The elements that form the polyatomic interferences (including C, Cl, H, N and O) usually result from the sample matrix, argon plasma, nebulized water and dissolved/entrained air. Most of the interference can be reduced by using a collision-reaction interface (CRI) or dynamic reaction cells, while other would require a sector field ICP-MS. The authors should specify for As, Hg and Cr which systems were used to control or eliminate interference during the measurements.

Response: We really appreciate the reviewer's comments.

Polyatomic interference is the main spectral interference source of ICP-MS, which is solved by collision reaction pool technology. And the octopole collision-reaction system was applied in Agilent 7700 ICP-MS to eliminate mass spectral interference, such as $^{35}\text{Cl}^{16}\text{OH}$ interference on ^{52}Cr , $^{40}\text{Ar}^{35}\text{Cl}$ interference on ^{75}As and so on.

Non-mass spectrum interference mainly includes matrix effect, space charge effect, and physical effect interference. The interference degree is related to the matrix property of the sample. The interference was eliminated and reduced by diluting the sample, internal standard method and optimizing instrument conditions.

In addition, The soil samples were digested by a mixed acid of nitric acid and hydrochloric acid on a microwave digestion system according to the USEPA method 3052 [1]. This mixed acid extraction, avoiding the addition of Cl⁻(HCl), has better analysis effect when compared with aquaregia extraction. And microwave digestion avoids the loss of As and Hg.

- Memory effects are major challenges in Hg determination by ICP-MS. How was it checked?

Response: Thanks for the reviewer's questions. The 0.18% L-cysteine was added into the sample solution to eliminate the memory effect of mercury[2], Now, this information was supplemented and inserted into the text accordingly (Lines 74-75).

- An internal standard (IS) is important for monitoring the change in the ICP-MS signal due to the plasma instability and blockage of the orifice of the sample cone, etc. Unfortunately, I found no information and discussion regarding the use of IS.

Response: We really appreciate the reviewer's comments. Now, this information was supplemented and inserted into the text accordingly (Lines 76-77) and detailed description were presented in supporting information as follows.

- 1) an internal standard solution of eight elements for ICP-MS analysis, included Li, Sc, Ge, Y, Rh, In, Re and Bi (CFGG-163175-02-01), was obtained from the National Standard Samples Website.
- 2) All samples were spiked with internal standard solution, the isotope of each element and its corresponding internal standard were selected and presented in Table SI [3].

Table S1 Isotopes of heavy metals and its corresponding internal standard for ICP-MS analysis

Element	Isotope	Internal standard
---------	---------	-------------------

As	75	⁷⁴ Ge
Cd	111	¹¹⁵ In
Cr	52	⁴⁵ Sc
Cu	63	⁷⁴ Ge
Hg ^a	202	¹¹⁵ In
Ni	60	⁷⁴ Ge
Pb	208	²⁰⁹ Bi
Zn	66	⁷⁴ Ge

^a: The data of Hg was obtained from a literature[2] and other elements were obtained from the National Environmental Protection Standards of China (HJ 803-2016) [3] .

- Significant figures must be corrected along the text and tables. Concentration values and standard deviation must be written to not more than three and two significant figures, respectively. Example: on page 3, line 20: 12.9 ± 4.83 mg/kg  12.9 ± 4.8 mg/kg; 89.2 ± 53.2 mg/kg  89 ± 53 mg/kg.

Response: As suggested by the reviewers, the significant figures have been extensively revised according to the suggestions (Line 20, Lines 111-112, and Table 1).

- On page 4, line 54: Xiangfen county

Response: Revised as suggested, this sentence “some studies have reported that high levels of metals from soil were detected in Shanxi, such as Xiangfen county [15], Xinzhou [16] and Fenhe basin [17]” has been rewritten as follow “some studies have reported that high levels of metals from soil were detected at Xiangfen county [15], Xinzhou area [16] and Fenhe basin [17] in Shanxi Province of China” (Lines 55-56).

References:

[1] USEPA 1996. SW-846 Test Method 3052: Microwave assisted acid digestion of siliceous and organically based matrices. U.S. Environmental Protection Agency, Washington, DC.

[2] Li Y. F., Chen C. Y., Li B., Sun J., Wang J. X., Gao Y. X., Zhao Y. L., Chai Z. F. 2006 Elimination efficiency of different reagents for the memory effect of mercury using ICP-MS. *J.*

Anal. At. Spectrom. **21**: 94-96. (10.1039/B511367A)

[3] NEPSC (National Environmental Protection Standards of China). 2016 Soil and sediment determination of aqua regia extracts of 12 metal elements-Inductively coupled plasma mass spectrometry (in Chinese). Beijing, China.

[4] Natasha, Shahid M., Khalid S., Bibi I., Bundschuh J., Niazi N. K., Dumat C. 2020 A critical review of mercury speciation, bioavailability, toxicity and detoxification in soil-plant environment: Ecotoxicology and health risk assessment. *Sci. Total Environ.* **711**. (10.1016/j.scitotenv.2019.134749)

[5] Zhu W., Li Z. G., Li P., Yu B., Lin C. J., Sommar J., Feng X. B. 2018 Re-emission of legacy mercury from soil adjacent to closed point sources of Hg emission. *Environ. Pollut.* **242**: 718-727. (10.1016/j.envpol.2018.07.002)

[6] Luo L., Ma Y., Zhang S., Wei D., Zhu Y. G. 2009 An inventory of trace element inputs to agricultural soils in China. *J. Environ. Manage.* **90**: 2524-30. (10.1016/j.jenvman.2009.01.011)

[7] CNEMC (China National Environmental Monitoring Center). 1990 Soil elements background values in China (in Chinese), China Environmental Science Press, Beijing.

[8] CSY (China Statistics Yearbook). 2019 Table 9-1: energy production in major years., <http://www.stats.gov.cn/tjsj/ndsj/2019/indexch.htm>.

[9] SSY (Shanxi Statistics Yearbook). 2018 Table 6-6: Coal production in major years. <http://tjj.shanxi.gov.cn/tjsj/tjnj/nj2018/indexch.htm>

[10] CSY (China Statistics Yearbook). 2019 Table 9-2: energy consumption in major years. <http://www.stats.gov.cn/tjsj/ndsj/2019/indexch.htm>.

[11] SSY (Shanxi Statistics Yearbook). 2018 Table 6-7: Coal consumption in major years. <http://tjj.shanxi.gov.cn/tjsj/tjnj/nj2018/indexch.htm>

[12] Gao X., Xu M., Hu Q., Wang Y. 2016 Leaching behavior of trace elements in coal spoils from Yangquan coal mine, northern China. *J Earth Sci* **27**: 891-900. (10.1007/s12583-016-0720-6)

[13] Zhu D., Wei Y., Zhao Y., Wang Q., Han J. 2018 Heavy Metal pollution and ecological risk assessment of the agriculture soil in Xunyang Mining Area, Shaanxi Province, Northwestern China. *Bull. Environ. Contam. Toxicol.* **101**: 178-184. (10.1007/s00128-018-2374-9)

[14] Li Z., Ma Z., van der Kuijp T. J., Yuan Z., Huang L. 2014 A review of soil heavy metal pollution from mines in China: pollution and health risk assessment. *Sci. Total Environ.* **468**: 843-53. (10.1016/j.scitotenv.2013.08.090)

[15] Pan L. B., Ma J., Wang X. L., Hou H. 2016 Heavy metals in soils from a typical county in Shanxi Province, China: Levels, sources and spatial distribution. *Chemosphere* **148**: 248-54. (10.1016/j.chemosphere.2015.12.049)

[16] Shangguan Y., Cheng B., Zhao L., Qin X., Hou H., Xu Y., Zhao R., Zhang Y., Hua X., Huo X., Zhao X. 2017 Occurrences, distributions, and multivariate analyses of trace elements in agricultural soils in the Xinzhou area of Shanxi, China. *Pedosphere* **28**: 542-554. (10.1016/s1002-0160(17)60304-7)

[17] Liu M. X., Han Z. Q., Yang Y. Y. 2019 Accumulation, temporal variation, source apportionment and risk assessment of heavy metals in agricultural soils from the middle reaches of Fenhe River basin, North China. *RSC Adv.* **9**: 21893-21902. (10.1039/c9ra03479j)

Appendix B

Aug 16, 2020

Dear Dr. Laura Smith,

Thank you very much for your favorable consideration and for the insightful comments of RSC Associate Editor and reviewer 2 concerning our manuscript entitled “Heavy metals contamination and ecological risk assessment of the agricultural soil in Shanxi Province, China” (RSOS-200538). Those comments are very helpful for improving our paper. Here below is our description on the revisions according to the Associate Editor and reviewer 2. We list all the changes in the response as follows. We hope you find the revised manuscript suitable for publication in *Royal Society Open Science*.

Once again, thank you very much for your time and consideration.

Sincerely,

Lihong Li

lilh@jzxy.edu.cn

Responses to the comments:

RSC Associate Editor:

Thank you very much for the effort to improve this manuscript. A few more issues need to be addressed before your manuscript is ready for publicación. Please address the comments for Reviewer 2 and the following: The format of the abstract must meet the standard requirements of objectives, materials and methods, results and conclusions. In its current version, the materials and methods section are incomplete. Additionally, a professional proofreading service is highly recommended as the language needs a trough revision.

Response: We really appreciate the RSC Associate Editor’s positive comments. The materials

and methods section in abstract have been revised according to the suggestions and expression as follows: “a total of 33 samples in the surface soil were collected from 11 cities in Shanxi. The soil samples were digested by a mixed acid of nitric acid and hydrofluoric acid on a microwave digestion system, then the levels of eight heavy metals were analyzed using inductively coupled plasma mass spectrometer. The pollution levels of soil heavy metals were evaluated using geo-accumulation index and their ecological risks were assessed using risk index calculated by Hakanson’s method..” (Lines 17-22)

The original statements as follows: “a total of 33 samples in the surface soil were collected from 11 cities, and the levels of eight heavy metals were analyzed using inductively coupled plasma mass spectrometer.”

Reviewer 2

Compared to the first draft, the authors have considerably improved the manuscript. Most of the comments have been taken into account. It is my opinion that the manuscript is worthy of publication in Royal Society Open Science after addressing the following issues:

- Regarding a response to the comments, I would like to highlight that the USEPA method 3052 has several additional alternative acid and reagent combinations including hydrochloric acid and hydrogen peroxide and only the reagent mixture HNO₃/H₂O₂ (or HNO₃/HF) avoid the addition of Cl (HCl).

- In the “Supporting Information”, the authors report the following digestion method: “The digestion was carried out on a microwave digestion system (WX-6000, PreeKem Scientific Instruments Co., Ltd.) with a digestion solution of 7 ml of nitric acid and 3 ml of hydrofluoric acid”; while on page 3, lines 70-71 of the revised manuscript “The soil samples

were digested by a mixed acid of nitric acid and hydrochloric acid on a microwave digestion system according to the USEPA method 3052”. Which reagent mixture for microwave assisted digestion has been used? HNO₃/HCl or HNO₃/HF?

Response: We really appreciate the reviewer's comment. The soil samples were digested by the reagent mixture HNO₃/HF to avoid the addition of Cl (HCl). And the corresponding description were presented at the text (Lines 72-73).

- Supporting Information: Table S1: the Hg data is not present;

Response: We are sorry for this negligence. The Hg data was supplemented in Table S1, which have adjusted to the Table S5 now.

on page S3 line 47 “The detect limitation for eight metals were 0.011 – 0.038 mg/kg dw” should be “The detection limits for eight metals were in the range of 0.011 – 0.038 mg/kg dw”;

Response: As suggested by the reviewers (Line 63 on page S8).

line 39: 97.6% – 106.2% should be 97.6 – 106%; 78% – 129% should be 78 – 129%. etc.

Response: As suggested by the reviewers (Line 66 on page S8).

- Abstract: Significant figures must still be corrected along the text, tables and “Supporting Information”: 43.4 ± 14 mg/kg should be 43 ± 14 mg/kg, 27.3 ± 19 mg/kg should be 27 ± 19 mg/kg; 17.0 ± 13 mg/kg should be 17 ± 13 mg/kg, and 89.2 ± 53 mg/kg should be 89 ± 53; lines 48-50: 26.63% (25.21%-29.02%) should be 26.6% (25.2-29.0%) etc....

Response: The significant figures have been extensively revised according to the suggestions (Lines 23-24, Line 51, Line 53, Lines 113-114, Table 1 and Table S3).

- On page 3, line 50: Table S2: Number tables consecutively in accordance with their appearance in the text (Table S1, S2 etc).

Response: We thank the reviewer for the constructive suggestion. The Number tables in the SI have been rearranged with their appearance in the text.